# COVID-19 Patients in the COVID-19 Recovery and Engagement (CORE) Clinics in the Bronx

**DOI:** 10.3390/diagnostics13010119

**Published:** 2022-12-30

**Authors:** Anna Eligulashvili, Megan Darrell, Carolyn Miller, Jeylin Lee, Seth Congdon, Jimmy S. Lee, Kevin Hsu, Judy Yee, Wei Hou, Marjan Islam, Tim Q. Duong

**Affiliations:** 1Department of Radiology, Montefiore Medical Center, Albert Einstein College of Medicine, Bronx, NY 10461, USA; 2Department of Medicine, Montefiore Medical Center, Albert Einstein College of Medicine, Bronx, NY 10461, USA; 3Department of Family, Population and Preventive Medicine, Renaissance School of Medicine at Stony Brook University, Stony Brook, NY 11794, USA

**Keywords:** PASC, long COVID, pulmonary function tests, brain imaging, chest imaging, COVID symptoms, fatigue, shortness of breath

## Abstract

Background: Early in the pandemic, we established COVID-19 Recovery and Engagement (CORE) Clinics in the Bronx and implemented a detailed evaluation protocol to assess physical, emotional, and cognitive function, pulmonary function tests, and imaging for COVID-19 survivors. Here, we report our findings up to five months post-acute COVID-19. Methods: Main outcomes and measures included pulmonary function tests, imaging tests, and a battery of symptom, physical, emotional, and cognitive assessments 5 months post-acute COVID-19. Findings: Dyspnea, fatigue, decreased exercise tolerance, brain fog, and shortness of breath were the most common symptoms but there were generally no significant differences between hospitalized and non-hospitalized cohorts (*p* > 0.05). Many patients had abnormal physical, emotional, and cognitive scores, but most functioned independently; there were no significant differences between hospitalized and non-hospitalized cohorts (*p* > 0.05). Six-minute walk tests, lung ultrasound, and diaphragm excursion were abnormal but only in the hospitalized cohort. Pulmonary function tests showed moderately restrictive pulmonary function only in the hospitalized cohort but no obstructive pulmonary function. Newly detected major neurological events, microvascular disease, atrophy, and white-matter changes were rare, but lung opacity and fibrosis-like findings were common after acute COVID-19. Interpretation: Many COVID-19 survivors experienced moderately restrictive pulmonary function, and significant symptoms across the physical, emotional, and cognitive health domains. Newly detected brain imaging abnormalities were rare, but lung imaging abnormalities were common. This study provides insights into post-acute sequelae following SARS-CoV-2 infection in neurological and pulmonary systems which may be used to support at-risk patients and develop effective screening methods and interventions.

## 1. Introduction

Many survivors of coronavirus disease 2019 (COVID-19) experience lingering neurological and pulmonary symptoms that persist long after severe acute respiratory syndrome coronavirus 2 (SARS-CoV-2) infection has resolved [1,2,3,4]. These symptoms are commonly referred to as post-acute sequelae of SARS-CoV-2 infection (PASC). Neuro-PASC include, but are not limited to, altered mental status, anxiety, depression, dizziness, headaches, memory loss, and post-traumatic stress disorder [5,6,7,8]. Pulmonary PASC include shortness of breath, persistent cough, and fatigue. Neuro-PASC could be caused by direct infection of the central nervous system, cytokine storm, systematic illness secondary to the initial viral infection, or psychological stressors such as social isolation, stigma, and future uncertainty [9]. Pulmonary-PASC could be caused by direct infection of the pulmonary system and/or systematic illness secondary to the initial viral infection, including pneumonia, acute respiratory distress, hypoxia, sepsis, pro-inflammatory mediators, cytokine storm and other host-mediated immunological responses [10,11,12]. Individuals with mild symptoms from SARS-CoV-2 infection (i.e., not requiring hospitalization) may also be susceptible to neurological and pulmonary PASC [13,14].

New York City was hit hard by the first wave of COVID-19 and by multiple subsequent surges of infection from different variants [15]. The Montefiore Health System consists of 15 hospitals located in the New York Metropolitan area in the Bronx and its environs, serving a large and diverse patient population, including many patients with lower socioeconomic status. Early in the pandemic, we established two COVID-19 Recovery and Engagement (CORE) Clinics for COVID-19 survivors with protracted symptoms. A detailed evaluation protocol was implemented to assess physical, emotional, and cognitive function, including pulmonary function tests and post-COVID imaging findings. In this study, we report a detailed analysis of a subset of this CORE patient cohort that had pulmonary function tests, imaging tests, and a battery of symptom, physical, emotional, and cognitive assessments.

## 2. Methods

### 2.1. CORE Patients

This study was approved by the Einstein IRB (IRB# 2021-13658) with a waiver of informed consent and followed all relevant regulatory guidelines. This is a prospective observational study of COVID-19 adult patients with protracted symptoms who were referred to Montefiore Medical Center’s COVID-19 Recovery and Engagement (CORE) clinics between 26 June 2020 and 7 January 2022. Eligible patients were adults (≥18 years old) who had probable or confirmed COVID-19 and were experiencing new or continued symptoms 4 or more weeks later. Patients who were terminally ill, referred to hospice or interested in comfort measures only were not eligible for referral to the CORE. On average, patients visited the CORE Clinics 133 ± 108 days after COVID-19 diagnosis. There were 11 patients who returned time over 300 days. If these patients were to be removed, the standard deviation was 59.93 days.

Patients who had less than 70 answers to survey questions and clinical variables (out of 100) were excluded. This threshold albeit arbitrary was chosen to avoid using subjects with many missing data. The final cohort consisted of 97 CORE patients.

### 2.2. Demographics and Laboratory Data

Age, sex, race, ethnicity, hospitalization status, and critical illness information was collected through patient questionnaires. Data from electronic medical records (EMR) were extracted automatically as described previously [16,17,18]. Preexisting comorbidities included body mass index (BMI), congestive heart failure (CHF), chronic kidney disease (CKD), hypertension, chronic obstructive pulmonary disease (COPD) and asthma that were designated by ICD10 codes at admission or prior. Hospitalization status and intensive-care-unit (ICU) admission were also extracted.

### 2.3. Symptom, Physical, Emotional, and Cognitive Assessments

All patients were surveyed for the presence of various post-infection symptoms. They were additionally administered a Modified Edmonton Physical Symptom Assessment (MEPSA) [19]. The original Edmonton Physical Symptom Assessment, initially validated in advanced cancer patients, included a list of common symptoms and allowed for repeated quantitative measurement of symptom intensity with minimal patient burden [20]. In our MEPSA, we asked patients to quantify their level of 14 symptoms over the past week prior to presentation to the clinic using a numerical rating scale between 0 (no symptom or best overall wellbeing) and 10 (the highest level of symptom imaginable or worst overall wellbeing).

Assessments of physical, emotional, and cognitive functions were performed. Disability and functional status were assessed using a modified Katz Activities of Daily Living (ADL) [21], Instrumental Activities of Daily Living (IADL) [22], and markers of frailty [23]. We assessed for depression, anxiety and post-traumatic symptoms using the Patient Health Questionnaire (PHQ-9) [24], Generalized Anxiety Disorder (GAD-7) [25], and Post-Traumatic Stress List for DSM-5 (PCL-5) questionnaires, respectively. We used previously validated cut-points for depression (PHQ ≥ 10) [26,27], anxiety (GAD-7 ≥ 8) [25,28] and post-traumatic symptoms (PCL-5 > 33) [29]. Social determinants of health (SDOH) information was collected through patient questionnaires.

### 2.4. Pulmonary Assessments

During the clinic visit (which could be remote or in-person), patients were evaluated by either general internal medicine or a pulmonary/critical care physician. The visits included a debrief of the acute illness experience, a summary of the questionnaire scores, an exploration of the patients’ persistent symptoms, a physical examination (for in-person visits), medication reconciliation, and interventions based on the clinical assessment. We used chart review to collect additional information on the patient’s history and demographics and to summarize relevant clinical interventions. Clinical measures of physical pulmonary health included the Borg Dyspnea Scale at rest and exertion, Lung Ultrasound (LUS) Score, and triplicate Diaphragm Excursion measurements with quiet and deep inspiration for survivors of critically ill COVID-19 pneumonia, defined as hypoxemia requiring high-flow nasal cannula or invasive or non-invasive positive pressure ventilation. All LUS videos were independently reviewed and scored by a blinded reviewer trained in scoring Lung Ultrasound Scores. Patients also underwent spirometry testing, including the Six Minute Walk Test (6MWT) and Pulmonary Function Test (PFT), which provided measures of distance, maximum heart rate (HR), and percent oxygen saturation (SpO_2_), as well as forced vital capacity (FVC % Predicted), forced expiratory volume (FEV1 % Predicted), the FEV1/FVC ratio, the respiratory volume to total lung capacity (RV/TLC % Predicted) ratio, and the diffusing capacity of lung carbon monoxide (DLCO % Predicted), respectively [30].

### 2.5. Lung Ultrasound Protocol

For the LUS, subjects were scanned in the upright position. Twelve fields were assessed (six zones per hemithorax). Each zone was scanned based on predetermined anatomical landmarks to assess anterior, medial, and posterior lung areas per published protocols [31,32].

LUS was scored by identification of four sonographic patterns of lung ultrasound [31,32]: normal lung by the presence of lung sliding with A-lines (score 0); the presence of significant B-lines (score 1); confluent B-lines with or without subpleural consolidations (score 2); and extensive subpleural consolidations with B-lines (score 3). The total lung ultrasound score was calculated by a composite of the sum of all individual zone scores, ranging between 0 and 36 [33].

### 2.6. Diaphragm Ultrasound Protocol

Subjects underwent scanning in a supine position of the right hemidiaphragm via a subcostal approach, using two-dimensional B-mode and M-mode ultrasound views [34]. Diaphragm excursion was measured during resting respiration (quiet breathing) and deep inspiration (deep breathing) [34]. We obtained (3) consecutive measurements which were averaged into a final composite distance recording in centimeters (cm).

### 2.7. Imaging Assessments

Images and radiology reports of chest x-ray, chest computed tomography (CT) images, head CT, and brain magnetic resonance imaging (MRI) were evaluated at three time points: pre-COVID-19, during COVID-19, and post-COVID-19, if available. Pre-COVID-19 images were at least one month prior to COVID-19 diagnosis. During-COVID images included the first image taken after COVID-19 diagnosis. Post-COVID images included the first images taken at least one month after discharge from COVID-19 hospitalization. For patients who were not hospitalized for acute COVID-19 infection, there were generally no during-COVID-19 images.

For brain imaging data, two board-certified neuroradiologists with 16 and 11 years of experience evaluated the brain images and radiology reports in a single setting and consensus was reached for all assessments. The primary findings on brain imaging included new and prior stroke and hemorrhage. Secondary findings included microvascular disease (MVD), atrophy, and white matter (WM) changes. The presence or changes (for patients with multiple time point data) in stroke and hemorrhage, MVD, volume loss, and WM (i.e., hyperintensity or lesion) were noted with qualitative adjustment for age (i.e., whether brain volume loss is appropriate for patient’s age). Mass effect (from tumor or infarction) if any was also noted.

For chest imaging data, radiology reports were evaluated for the presence of unilateral or bilateral lung opacity or consolidation/infiltrate and the presence of fibrosis-like changes in the lung at the three time points, if available, were tabulated.

### 2.8. Statistical Analysis

Analysis of group differences for categorical variables employed χ^2^ tests and for categorical variables student *t*-tests via the statistical library from SciPy Python package in JupyterLab (©2018, Project Jupyter, https://jupyter.org, accessed on 25 December 2022). Sensitivity power analysis was performed. Cramer’s V and Cohen’s D values for size effect investigation were computed. *p* < 0.05 was considered statistically significant unless specified otherwise.

## 3. Results

### 3.1. Patient Demographics

Table 1 shows the profiles of the CORE patients. The entire cohort was 58.61 ± 14.43 years old, 49.48% female, 49.48% Hispanic, and 26.80% Black Non-Hispanic, with a BMI of 32.61 ± 7.63. Prevalence of diabetes was 32.0%, hypertension 56.7%, COPD/asthma 38.1%, CHF 9.3% and CKD 9.3%. Most patients were hospitalized (77/97) and about half (50) were critically ill. Effect size measures across all variables in Table 1 were small. Age was the only variable that was significantly different between hospitalized and non-hospitalized cohort (*p* < 0.05). For the sample size of 20 non-hospitalized and 77 hospitalized patients, the minimum detectable effect sizes for the 80% and 90% statistical power were 0.71 and 0.82, respectively.

### 3.2. Symptom, Physical, Emotional, and Cognitive Assessments

Table 2 summarizes the symptom, MEPSA, ADL, IADL, frailty, PHQ-9, GAD-7 and PCL-5 assessments. For all patients, dyspnea (77.32%), fatigue (70.1%), decreased exercise tolerance (55.67%), and brain fog/cognitive issues (40.21%) were the most reported symptoms. Many CORE patients reported fatigue (3.87 ± 3.23), overall well-being score (3.45 ± 3.14), and shortness of breath (3.32 ± 2.96). The average ADL score was 0.57 ± 1.14 (out of 6) and IADL was 2.11 ± 2.53 (out of 8), indicative of high function but some dependency. Challenges with bathing (16.84%) and dressing (13.54%) were the most reported ADLs and help with shopping (41.24%) and laundry (36.84%) were the most reported IADLs. The average frailty score was 1.66 ± 1.4 (out of 5), indicating pre-frailty. Of the behavioral health surveys, the GAD-7 anxiety score was 3.8 ± 5.43, PCL-5 PTSD score was 13.12 ± 14.9, and PHQ-9 depression score was 5.64 ± 6.08. Few patients reported SDOH challenges (0.40 ± 0.79 out of 8) and safety concerns (0.1 ± 0.31 out of 2).

These test scores were not significantly different between hospitalized and non-hospitalized cohort after multiple comparison correction (*p* > 0.05), except GAD-7 score which was higher in hospitalized compared to the non-hospitalized group (4.23 ± 5.94 vs. 2.15 ± 2.06, *p* = 0.039). The difference between the hospitalized and non-hospitalized groups are in the range of small to medium effect sizes in Table 2.

### 3.3. Pulmonary Assessments

Table 3 summarizes the 6MWT, Borg dyspnea scale, LUS, diaphragm excursion, and PFT results for the hospitalized and non-hospitalized cohorts. The 6MWT distance and 6MWT maximum heart rate were below normative values but were not significantly different between hospitalized and non-hospitalized cohorts (*p* > 0.05). 6MWT SpO_2_% was abnormal for the hospitalized cohort (89 ± 7%) but normal (95 ± 1.8%) for the non-hospitalized cohort with a significant difference between groups (*p* < 0.001).

The Borg dyspnea data indicated patients had slight shortness of breath at rest and moderate difficulty breathing during exertion, but there were no significant differences between groups (*p* > 0.05). The mean LUS score was abnormal for hospitalized patients but normal for non-hospitalized patients, with a significant difference between groups (*p* < 0.001). The average diaphragm excursion measurement with quiet inspiration was significantly higher in hospitalized patients (*p* < 0.05); however, the average diaphragm excursion measurement with deep inspiration was not significantly different between groups (*p* > 0.05).

FVC% Predicted, FEV1% Predicted, RV/TLC% Predicted, and DLCO% Predicted were abnormal for the hospitalized cohort, but normal for the non-hospitalized cohort, with significant differences between groups (*p* < 0.031, 0.04, 0.010, and 0.018, respectively). FEV1/FVC was normal for both groups and was not significantly different between groups (*p* > 0.05). Across all variables, effect size measures ranged from small to large in Table 3.

### 3.4. Imaging Assessments

Table 4 summarizes the brain imaging findings pre-, during and post-COVID-19. The average follow-up imaging was 5 months post-COVID-19 diagnosis. Many but not all patients (N = 31, 6, and 13 out of 97, for pre-, during and post-COVID-19, respectively) had clinically indicated brain MRI or head CT.

Four patients had prior stroke, one patient had newly detected hemorrhage during COVID-19 hospitalization, and 2 patients had newly detected hemorrhage post-COVID. Overall, newly detected major neurological events post-COVID-19 were rare.

Although the presence of MVD was frequently noted, it was consistent with patient’s age. Newly detected MVD was rare post COVID-19. Similarly, some patients had brain volume loss (atrophy) consistent with their age and newly detected atrophy was also rare post-COVID-19. For a patient subgroup who had both pre- and post-COVID-19 brain imaging (N = 6), 3 patients exhibited longitudinal changes (3 MVD, 1 atrophy, and 3 WM changes). The average time difference between the two time points was 7.06 years in this subgroup.

Table 5 summarizes the pulmonary imaging findings pre-, during- and post- COVID-19. The average follow-up scan was also 5 months post-COVID-19 diagnosis. Most patients had clinically indicated chest imaging performed (64, 75 and 70 out of 97 patients for pre, during and post-COVID-19, respectively). Opacity findings were rare pre-COVID-19 (N = 4), but very common during COVID-19 (N = 60), most were bilateral (N = 52). Most patients with post-COVID-19 lung imaging had opacity findings (N = 40), and most were also bilateral (N = 31). Fibrosis-like changes were rare pre- and during- COVID-19 (N = 5 and 1, respectively) but common post-COVID-19 (N = 14).

With respect to hospitalization status, lung opacity was more common in the hospitalized compared to the non-hospitalized group both during- and post-COVID-19 (*p* < 0.05, Appendix A). Fibrosis-like changes were more common in the hospitalized cohort post-COVID-19 (*p* < 0.05).

## 4. Discussion

This study characterized a subset of the COVID-19 CORE clinic patients. The major findings are: (i) dyspnea, fatigue, decreased exercise tolerance, brain fog, and shortness of breath were the most common symptoms, (ii) some patients have abnormal physical, emotional, and cognitive scores but most are able to live and function independently, (iii) some variables of the 6MWT, LUS, and diaphragm excursion are abnormal in the hospitalized cohort, suggestive of interstitial lung disease, (iv) pulmonary function tests suggest moderately restrictive pulmonary function in the hospitalized cohort but no obstructive pulmonary function nor air trapping in the lung, (v) newly detected major neurological events, microvascular disease, atrophy, and WM changes are rare, and (vi) many patients have persistent lung opacity and fibrosis-like findings post-COVID-19.

### 4.1. Symptom, Physical, Emotional, and Cognitive Assessments

The high incidence of dyspnea, fatigue, brain fog, decreased exercise tolerance, and shortness of breath are consistent with the literature. Goertz et al. found that fatigue, dyspnea, headache, and chest tightness were the four most common persistent symptoms across both hospitalized and non-hospitalized COVID-19 patients 3 months after infection [37]. Huang et al. reported that fatigue or muscle weakness was by far the most common symptom in hospitalized COVID-19 patients at 6 months followed up [38]. Carfi et al. found fatigue, dyspnea, and joint pains to be the three most prevalent symptoms in COVID survivors about 2 months after their COVID-19 hospitalization [39]. These symptomologies are not surprising given many COVID-19 patients were often discharged with major medical referrals [40,41].

Collectively, ADL, IADL, and frailty marker scores indicate that CORE patients have a mild dependency and prefrailty. Although CORE patients showed signs of anxiety, depression, and PTSD, they were below the clinical cutoff in both hospitalized and non-hospitalized patients. Other studies reported clinically significant neurological and psychiatric symptoms after COVID-19 diagnosis [42].

Surprisingly, there were few differences in symptom, physical, emotional, and cognitive scores between hospitalized and non-hospitalized cohorts. A likely explanation is that non-hospitalized COVID-19 patients who visited our CORE clinics likely had more symptomatic COVID-19 disease. Previously studies have reported individuals with mild symptoms from SARS-CoV-2 infection (i.e., not requiring hospitalization) may also be susceptible to neuro-PASC [13,14].

### 4.2. Pulmonary Assessments

For most pulmonary measures, only the hospitalized cohort exhibited abnormalities. Both 6MWT distance and maximum heart rates were below normal (compared to some normative means) because patients as a cohort were not able to accomplish the task. 6MWT and SpO_2_ saturation were clearly abnormal in the hospitalized cohort. Note that the normative data used for reference were from a very different population that were not matched (i.e., for age or sex, etc.) and thus comparisons need to be interpreted with caution.

LUS has been previously used to measure the degree of loss of aeration in patients with acute respiratory distress syndrome, with higher scores indicating more loss of normal lung aeration. B-lines in lung ultrasound denote artifacts reflecting a widening of the interlobular septa of the secondary pulmonary lobule. As a result, this finding is non-specific, and acute respiratory failure may represent cardiogenic or non-cardiogenic pulmonary edema as in ARDS, or other diagnoses [43]. In the post-COVID-19 population, however, high LUS likely denotes the presence of interstitial lung disease and lung injury following severe viral pneumonia. The abnormalities in spirometry measurements are suggestive of residual lung disease likely from COVID-19 pneumonia. Our spirometry analysis did not account for lung volume differences among patients thus such comparisons need to be interpreted with caution.

A combination of moderately reduced FVC%, FEV1%, RV/TLC ratio and DLCO% were abnormal in the hospitalized cohort, suggesting restrictive pulmonary function and air-trapping. FEV1/FVC in the hospitalized cohort was however normal, suggesting there was no significant obstructive pulmonary function. A normal FEV1/FVC ratio with a decreased FVC indicates a restrictive lung condition, which includes pulmonary fibrosis and infections such as pneumonia. A decreased FEV1/FVC ratio indicates an obstructive condition, such as asthma or COPD. In contrast, all these PFT variables were normal in the non-hospitalized cohort.

### 4.3. Imaging Findings

Although the sample size for brain imaging is small and both brain MRI and CT were included (which provide different sensitivity to detection of abnormalities), the main findings will likely hold, namely, that newly detected major neurological events post-COVID-19 are rare, and newly detected MVD, atrophy, and WM changes are also rare 5 months post -COVID-19 in our cohort. We thus concluded that there is no evidence of widespread changes in routine clinical brain imaging in our CORE patients 5 months post-COVID-19. However, it is possible that subtle brain changes existed, but were not detectable by routine clinical imaging. Imaging studies using more advanced imaging methods (such as diffusion tensor imaging, quantitative susceptibility mapping, and functional MRI) are warranted. It is surprising that given the broad spectrum of neuro-PASC symptoms reported in this cohort, there were comparatively few observable radiological abnormalities post-COVID-19. It is possible that these symptoms and neurological abnormalities have not yet manifested into structural changes in the brain on routine clinical imaging methods, and thus longer follow-up and more sophisticated imaging tools are necessary.

A few studies have reported brain imaging findings in post-COVID-19 patients. A cohort study found a greater reduction in grey matter thickness and tissue contrast in the orbitofrontal cortex and parahippocampal gyrus, greater changes in markers of tissue damage in regions that are functionally connected to the primary olfactory cortex and a greater reduction in global brain size in the COVID-19 cases compared to negative controls [44]. Abnormalities and cerebral microstructural changes in the brain of COVID-19 survivors both with and without neurological manifestations were noted [45,46], and persistent WM changes and ischemic stroke were associated with COVID-19 [47,48]. Note that many published studies to date were case reports or did not have pre-COVID-19 imaging data or controls (including the current study), which makes it difficult to definitively discern whether imaging abnormalities were pre-existing or a consequence of COVID-19 disease. Thus, there is likely reporting bias of positive clinical imaging findings associated with COVID-19. Brain imaging studies with proper controls with a correlation of neurological function at longer follow-up intervals are needed.

In contrast to brain imaging findings, there are clear anatomical abnormalities in the lung 5 months post-COVID-19. It is concerning that lung opacity in many patients has not completely resolved and that many patients developed pulmonary fibrosis-like changes 5 months after COVID-19. Pre-COVID-19 abnormalities were very rare and thus essentially all new lung findings were due to COVID-19. Hospitalized patients had more opacity and fibrosis-like changes post-COVID-19, consistent with disease severity. The incidence of fibrosis-like changes is likely underestimated in our study because a chest radiograph was included which has lower sensitivity for fibrosis detection compared to CT.

A few studies have previously reported persistent fibrosis-like lung changes post-COVID-19. CT abnormalities were common at 3 months after COVID-19 but with signs of fibrosis in a minority. More severe acute disease was linked with CT abnormalities at 3 months [49]. One study reported that although COVID-19 survivors showed continuous improvement in chest CT, residual lesions could still be observed and correlated with lung volume parameters one-year post-COVID-19 and the risk of developing residual CT opacities increases with age [50]. Another study found a significant percentage of individuals develop pulmonary sequelae after COVID-19 pneumonia, regardless of the severity of the acute process [51]. Six-month follow-up CT showed fibrotic-like changes in the lung in more than one-third of patients who survived severe COVID-19 pneumonia. These changes were associated with older age, acute respiratory distress syndrome, longer hospital stays, tachycardia, noninvasive mechanical ventilation, and higher initial chest CT score [52]. Given the numerous reports of persistent post-COVID-19 pulmonary sequela in many COVID-19 patients, longitudinal monitoring by chest imaging and pulmonary function in at-risk patients is warranted. Moreover, SARS-CoV-2 infection could also worsen existing pulmonary diseases. Many COVID-19 survivors are already being treated with pulmonary medications and pulmonary rehabilitation for pulmonary sequela.

### 4.4. Limitations

This study has several limitations. Our findings were limited to COVID-19 survivors who came to our CORE clinics and who were more likely to have more severe COVID-19 symptoms, and thus our cohort was not representative of the general population. It was not possible to definitively distinguish abnormalities that were due to COVID-19, pre-existing or worsened by COVID-19 disease although attempts were made to evaluate patients’ pre-pandemic data. SARS-CoV-2 infection often resulted in multi-organ injury and future studies should also investigate long COVID regarding multi-organ injury [17,18,40,41,53].

Sample sizes of imaging data were small and consisted of a mixture of imaging modalities with different sensitivities to pathology, and thus results must be interpreted with caution. The small sample size also precluded quantitative statistical parametric analysis of imaging data. Large multicenter longitudinal imaging studies with proper controls are needed. We hope to be able to report longer follow-up findings on this cohort in the future.

## 5. Conclusions

We established early in the pandemic the CORE Clinics for COVID-19 survivors. Many CORE patients experienced significant symptoms across the physical, emotional, and cognitive health domains. Pulmonary function tests suggest moderately restrictive pulmonary function in patients hospitalized with COVID-19. Newly detected major neurological events, microvascular disease, atrophy, and white-matter changes were rare, but persistent lung opacity and COVID-19-related lung fibrosis-like findings were common. Our study provides insights into neurological and pulmonary COVID-19 sequela which may be used to support at-risk patients and develop effective screening methods and interventions to address the potentially high burden of care needed among COVID-19 survivors.

## Figures and Tables

**Table 1 diagnostics-13-00119-t001:** Demographics of CORE patients. Mean ± SD or (%), N = 97. * *p* < 0.05 between hospitalized and non-hospitalized cohort.

	All (N = 97)	Hospitalized (N = 77)	Non-Hospitalized (N = 20)	Cohen’s D	Cramer’s V
Age (years)	58.61 ± 14.43	60.36 ± 13.53	50.8 ± 14.76 *	0.71	
Female	48 (49.48%)	32 (41.56%)	16 (80.00%)		0.311
Race/Ethnicity					
Hispanic	48 (49.48%)	42 (54.55%)	6 (30.00%)		0.199
Black Non-Hispanic	26 (26.8%)	16 (20.78%)	9 (45.00%)		0.224
White Non-Hispanic	13 (13.4%)	10 (12.99%)	3 (15.00%)		0.024
Other/Unknown	11 (11.34%)	9 (11.70%)	2 (10.00%)		0.022
BMI	32.61 ± 7.63	32.77 ± 7.79	32.01 ± 6.74	0.10	
Comorbidities					
CHF	9 (9.3%)	8 (10.39%)	1 (5.00%)		0.075
CKD	9 (9.3%)	8 (10.39%)	1 (5.00%)		0.075
Hypertension	55 (56.7%)	45 (58.44%)	10 (50.00%)		0.069
COPD/Asthma	37 (38.1%)	33 (42.86%)	8 (40.00%)		0.197
Diabetes	31 (32.0%)	29 (37.66%)	2 (10.00%)		0.240
Hospitalized	77 (79.38%)	na	na		
Critically Ill (IMV/ICU)	50 (51.55%)	50 (64.94%)	na		

**Table 2 diagnostics-13-00119-t002:** Symptoms, MEPSA, ADL, IADL, frailty and neuropsychiatric test scores of CORE patients (mean ± SD or N (%), N = 97). Continuous variables: *t*-test; Categorical variables: chi-squared (>10), Fisher’s test (<10). *p* (Bonf corr) = Bonferroni correction for multiple comparison. * indicates statistical difference.

	Variables	N	All Patients (N = 97)	Hospitalized (N = 77, 79%)	Non-Hospitalized (N = 20, 21%)	*p*	*p* (Bonf. Corr)	Cohen’s D	Cramer’s V
Presence of Symptoms	Dyspnea	97	75 (77.32%)	60 (77.9%)	15 (75%)	1.000	1		0.028
	Fatigue	97	68 (70.1%)	53 (68.8%)	15 (75%)	0.977	1		0.055
	Decreased Exercise Tolerance	97	54 (55.67%)	43 (55.8%)	11 (55%)	1.000	1		0.007
	Brain Fog/Cognitive Issues	97	39 (40.21%)	30 (39%)	9 (45%)	0.818	1		0.050
	Cough	97	23 (23.71%)	16 (20.8%)	7 (35%)	0.404	1		0.135
	Palpitations	97	15 (15.46%)	11 (14.3%)	4 (20%)	0.735	1		0.064
	Chest Pain	97	10 (10.31%)	7 (9.1%)	3 (15%)	0.445	1		0.079
	Abnormal Smell/Taste	97	13 (13.4%)	9 (11.7%)	4 (20%)	0.475	1		0.099
	Joint Pain	97	16 (16.49%)	13 (16.9%)	3 (15%)	1.000	1		0.021
	Lightheadedness/Dizziness	97	8 (8.25%)	6 (7.8%)	2 (10%)	0.672	1		0.032
	Symptoms Resolved	97	6 (6.19%)	6 (7.8%)	0 (0%)	0.594	1		0.131
	Post-Exertional Malaise	97	4 (4.12%)	1 (1.3%) *	3 (15%) *	0.036	0.504		0.279
	Headache	97	4 (4.12%)	4 (5.2%)	0 (0%)	0.582	1		0.106
	Back Pain	97	2 (2.06%)	2 (2.6%)	0 (0%)	1.000	1		0.074
MEPSA Questions Out of 10	Q2: Fatigue	97	3.87 ± 3.23	3.68 ± 3.25	4.6 ± 3.1	0.249	1	0.29	
	Q14: Overall Well-Being	97	3.45 ± 3.14	3.29 ± 3.22	4.1 ± 2.83	0.273	1	0.26	
	Q8: Shortness of Breath	97	3.32 ± 2.96	3.25 ± 2.85	3.6 ± 3.41	0.673	1	0.12	
	Q1: Pain	97	2.43 ± 2.90	2.55 ± 2.89	2 ± 2.97	0.468	1	0.19	
	Q3: Drowsiness	96	2.20 ± 2.76	2.25 ± 2.87	2 ± 2.34	0.688	1	0.09	
	Q12: Anxiety	97	2.18 ± 3.08	2.29 ± 3.17	1.75 ± 2.73	0.455	1	0.17	
	Q11: Depression	97	1.82 ± 2.89	1.97 ± 3.08	1.25 ± 1.94	0.201	1	0.25	
	Q6: Lack of Appetite	97	0.86 ± 1.95	0.86 ± 1.98	0.85 ± 1.84	0.988	1	0.00	
	Q4: Nausea/vomiting	97	0.43 ± 1.34	0.44 ± 1.4	0.4 ± 1.1	0.888	1	0.00	
ADL (out of 6) = 0.57 (1.14) High function, independent	Q4: Requires help bathing/getting in or out of shower	95	16 (16.84%)	16 (20.8%)	0 (0%)	0.071	0.426		0.225
	Q2: Requires help getting completely dressed	96	13 (13.54%)	13 (16.9%)	0 (0%)	0.120	0.72		0.203
	Q1: Requires help walking across room	96	11 (11.46%)	11 (14.3%)	0 (0%)	0.211	1		0.185
	Q3: Requires help transferring from bed to chair	97	9 (9.28%)	9 (11.7%)	0 (0%)	0.203	1		0.163
	Q5: Requires help because of problems controlling bladder or bowel	96	5 (5.21%)	4 (5.2%)	1 (5%)	1.000	1		0.005
	Q6: Requires help feeding self	96	1 (1.04%)	1 (1.3%)	0 (0%)	1.000	1		0.053
IADL (out of 8) = 2.11 (2.53) High function, independent	Q6: Requires help taking care of shopping needs	97	40 (41.24%)	36 (46.8%)	4 (20%)	0.215	1		0.220
	Q4: Requires help with doing laundry	95	35 (36.84%)	30 (39%)	5 (25%)	0.464	1		0.127
	Q3: Requires help with household tasks like cleaning, doing the dishes, or making the bed	96	34 (35.42%)	28 (36.4%)	6 (30%)	0.807	1		0.058
	Q8: Requires help traveling outside of home	97	31 (31.96%)	30 (39%) *	1 (5%) *	0.025	0.2		0.295
	Q5: Requires help handling purchases or other financial matters	96	28 (29.17%)	24 (31.2%)	4 (20%)	0.590	1		0.104
	Q7: Requires help planning, preparing or serving meals	94	27 (28.72%)	26 (33.8%) *	1 (5%) *	0.043	0.344		0.273
	Q2: Requires help taking medicine at correct time and dosage	97	7 (7.22%)	7 (9.1%)	0 (0%)	0.341	1		0.142
	Q1: Requires help using telephone	97	3 (3.09%)	3 (3.9%)	0 (0%)	1.000	1		0.091
Frailty Markers (out of 5) = 1.66 (1.4) Pre-Frail	Q1: Cannot stand up from chair without using arms	97	62 (63.92%)	46 (59.7%)	16 (80%)	0.570	1		0.171
	Q3: Cannot do moderate activities	97	49 (50.52%)	40 (51.9%)	9 (45%)	0.829	1		0.056
	Q2: Cannot climb one flight of stairs	96	48 (50%)	40 (51.9%)	8 (40%)	0.659	1		0.078
	Q5: Poor vision	97	19 (19.59%)	17 (22.1%)	2 (10%)	0.522	1		0.123
	Q4: Hard of hearing	97	10 (10.31%)	8 (10.4%)	2 (10%)	1.000	1		0.005
Behavioral Score	PHQ-9 Score (Depression)	97	5.64 ± 6.08	5.96 ± 6.67	4.4 ± 2.58	0.106	0.318	0.26	
	GAD-7 Score (Anxiety)	97	3.8 ± 5.43	4.23 ± 5.94 *	2.15 ± 2.06 *	0.013	0.039	0.39	
	PCL-5 Score (PTSD)	97	13.12 ± 14.9	14.05 ± 16.12	9.55 ± 8.04	0.085	0.255	0.30	
SDOH	SDOH Challenges Present (out of 8)	67	0.40 ± 0.79	0.48 ± 0.85	0.13 ± 0.52	0.058	0.117	0.44	
	SDOH Safety Concerns Present (out of 2)	67	0.1 ± 0.31	0.1 ± 0.3	0.13 ± 0.35	0.713	1	0.12	

**Table 3 diagnostics-13-00119-t003:** 6MWT, Borg Dyspnea scale, LUS Score, Diaphragm Excursion Measurements and PFT results (Mean ± SD). * *p* < 0.05, ** *p* < 0.01, *** *p* < 0.001 (*t*-test).

Variable	N	Hospitalized (N = 77, 79%)	Non-Hospitalized (N = 20, 21%)	*p*-Value	Normal Values or Ranges	Cohen’s D
6MWT (Distance, feet)	59	985 ± 429	773 ± 258	0.091	1873 ± 295 [35]	0.51
6MWT (Maximum heart rate, bpm)	55	111 ± 15	101 ± 24	0.341	128 ± 18 [36]	0.58
6MWT (SpO_2_ % Nadir)	56	89 ± 7 ***	95 ± 1.8 ***	<0.001	95 ± 2 [35]	0.74
Borg Dyspnea Scale (Rest)	76	0.96 ± 1.47	0.82 + 1.23	0.734	0 (normal) to 10 (worst).	0.10
Borg Dyspnea Scale (Exertion)	62	2.93 ± 2.23	2.86 ± 2.19	0.939	0 (normal) to 10	0.03
LUS Score	53	8.43 ± 6.73 ***	0.5 ± 0.71 ***	<0.001	0 (normal) to 36	1.19
Diaphragm Excursion, quiet inspiration (cm)	47	2.02 ± 0.72 *	1.74 ± 0.02 *	0.014	Range: 1.9–9	0.40
Diaphragm Excursion, deep inspiration (cm)	47	4.72 ± 1.82	5.18 ± 0.39	0.307	Range: 1.9–9	0.25
Pulmonary Function Tests						
FVC%	42	66.34 ± 18.93 *	90.6 ± 17.29 *	0.031	>80	1.29
FEV1%	42	68.73 ± 19 **	92 ± 10 **	0.004	>80	1.24
FEV1/FVC	42	81.02 ± 13	81.8 ± 7.6	0.852	>70	0.06
RV/TLC	19	122.65 ± 26.63 **	105 ± 2 **	0.010	<120	0.94
DLCO%	20	62.33 ± 23.90 *	94 ± 9 *	0.018	75–140%	1.92

6MWT—6 min walk test; HR—heart rate; bpm—beats per minute; SpO_2_—oxygen saturation; LUS—lung ultrasound score; FVC—forced vital capacity; FEV1—forced expiratory volume in the first second. For diaphragm excursion measurements, lower values are more abnormal. Please note that the last columns that indicate typical normal ranges are for general reference only. They are not matched controls (i.e., for age, BMI, sex, etc.). Reference [35]—normative cohort comparison: 46% female, mean age of 58, BMI of 27, 5% non-white. Reference [36]—normative cohort comparison: 20–50 years old, 100% non-white.

**Table 4 diagnostics-13-00119-t004:** Major brain imaging findings pre-, during and post-COVID-19 diagnosis. Post-COVID-19 average 182 days after diagnosis. Note that each patient could have multiple radiological findings and thus the counts are larger than unique patient counts.

	Pre-COVID-19	During-COVID-19	Post-COVID-19
Total N with Imaging	31/97	6/97	13/97
No remarkable findings	20	2	4
Positive findings	11	4	9
Primary Outcomes			
Acute Stroke	1	0	0
Prior Stroke	3	1	1
Newly Detected Hemorrhage	0	1	2
Secondary Outcomes			
MVD	6	2	8
Atrophy	0	0	2
WM HI/Lesions	3	0	0

**Table 5 diagnostics-13-00119-t005:** Major lung imaging findings pre-, during and post-COVID-19 diagnosis. N reflects the number of unique patients. Post-COVID-19 average 5 months after diagnosis (N = 97).

	Pre-COVID-19	During-COVID-19	Post-COVID-19
Total N with imaging	64	75	70
No opacity	59	15	30
Opacity	4	60	40
Unilateral	2	8	9
Bilateral	2	52	31
No fibrosis	59	74	56
Fibrosis	5	1	14

## Data Availability

Upon reasonable request via corresponding author.

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
