# Peer review of "COVID-19 Patients in the COVID-19 Recovery and Engagement (CORE) Clinics in the Bronx"

_diagnostics, 2022, doi:10.3390/diagnostics13010119_

Round 1

Reviewer 1 Report

I’d like to thank the authors and the Editorial Board for the opportunity to review the article submitted to Diagnostics. The authors' manuscript refers to a very important topic, the functioning of COVID-19 patients.  I believe that it presents important results from the practical point of view, but it's not free of limitations. I think that after some minor changes, the submitted article could be published. Below I present my comments on the individual section of the authors’ manuscript.

  1. Authors described the results of 97 CORE patients, where N=20 were not hospitalised. From the statistical point of view, such a small sample does not provide sufficient statistical power of at least 1-Beta=0.80, which would allow any test calculations. If possible, an additional cross-sectional study would be recommended to strengthen conclusions drawn on the obtained sample. Due to the diagnostic nature of the authors’ study, it cannot be considered a limitation, since this setting differs largely from the typical cross-sectional study. Therefore I highly recommend that authors perform the sensitivity power analysis (see Lakens, 2022) and show what size of the effect could be shown as significant for the obtained sample.

  2. Authors write that patients visited the core clinics M=133 days after the diagnosis. The standard deviation for this mean is fairly large (SD=108). It shows that the obtained sample was not homogenous. I recommend that the authors refer to that matter in the limitations section and try to discuss why there was such a large difference in the average day of the visit.

  3. I highly recommend that authors supplement all results with the proper effect-size measures. P-values are highly related to the obtained sample size and it is recommended to discuss the results based on the size of the effect rather than its significance. Therefore, I recommend that: (1) authors calculate effect size measures for all used statistical comparisons (they can report Cramer’s v or Yule’s Phi for chi-squared analysis, and Cohen’s d values for t-test comparisons), and (2) discuss the results based on the obtained effect sizes rather than their significance. For such a small sample size (N=97) it is not uncommon to obtain a large number of non-significant results. Additionally, authors could also calculate the Bayesian versions of the used analyses and report the values of Bayes Factors in order to make up for the limitations of the p-value statistics.

Author Response

"This study was approved by the Einstein IRB (IRB# 2021-13658, approved 12/21/2021) with a waiver of informed consent and followed all relevant regulatory guidelines. "

Reviewer 2 Report

In this study, the authors characterized a cohort of patients with COVID-19 referred to the CORE clinics between June 2020 and January 2022. The authors implemented a comprehensive protocol and evaluated many functional (physical, emotional, and cognitive), clinical, and laboratory aspects, as well as pulmonary function tests and imaging tests 5 months after acute COVID-19.

The topic is not new and many findings confirm current evidence and do not add novelty (e.g. post-COVID 19 lung fibrosis) to the field. Additionally, stratification by hospitalization status is not optimal given the small sample size (n=97). Additionally, it is not clear how only 97 patients were recruited and why the authors selected only those responding to at least 70 questions. Motivate why you chose this threshold and not another one.

Finally, most relevant findings concerning neuro PASCs and pulmonary PASCs should be the central focus of the manuscript and not a simple part of it.  I suggest the authors to restructure the paper by stressing the importance of clinical-radiological dissociation in patients with long-term neurocognitive symptoms along with a more detailed description of pulmonary radiological and LUS features) and give less space to more general and less relevant features. With data presented this way, I feel that so much information is included and the readers could not easily catch the importance of most relevant findings. Also, the categorization by hospitalization status does not add so much information, and should be reconsidered, especially considering the small sample size of the study population. 

Author Response

(The authors gave the same response as above.)

Round 2

Reviewer 2 Report

There are still major concerns to be addressed

Round 3

Reviewer 2 Report

The authors addressed the reviewers comments. The manuscript is ready for publication.